# The influences of urbanization on breeding behavior of American bullfrog (*Aquarana catesbeiana*) in South Korea

Ji-A. Lee[1‡], Md Mizanur Rahman[2‡], Seung-Ju Cheon[1], Amaël Borzée[3], Ha-Cheol Sung[2]*

**1** Department of Biological Sciences and Biotechnology, Chonnam National University, Gwangju, South Korea, **2** Department of Biological Sciences, Institute of Sustainable Ecological Environment, Chonnam National University, Gwangju, South Korea, **3** Laboratory of Animal Behaviour and Conservation, College of Life Sciences, Nanjing Forestry University, Nanjing, People's Republic of China

‡ These authors share first authorship.
* shcol2002@jnu.ac.kr

## Abstract

Urbanized areas often exhibit high levels of anthropogenic noise, which can mask or interfere with animal communication signals, especially those that use sound to communicate, making it challenging for individuals to detect and interpret acoustic cues. While calling is crucial for anuran breeding and communication, the coping mechanisms of city dwelling and cosmopolitan species in urbanized environments remain understudied. Given that invasive species have higher environmental adaptability than native species (because of habitat specificity and environmental sensitivity), we studied the calling patterns of the invasive American bullfrog (*Aquarana catesbeiana*) in different levels of urbanized areas in South Korea. In our study, we found an early onset of calling activities in urbanized *A. catesbeiana* populations, which suggests a change induced by factors related to urbanization on breeding phenology. Additionally, urban populations show more intense diurnal calling activities but shorter breeding periods than non-urban populations. The results indicate water temperature and urban noise as the leading factors influencing calling activities in *A. catesbeiana*. Urbanization induced changes in breeding and calling activities might have facilitated *A. catesbeiana* to invade and establish populations outside their natural ecological niches. Thus, this study sheds light on the effect of urbanization on frog breeding activities and how an invasive species copes with modified environments in new areas.

## 1. Introduction

Urban areas are landscapes where peoples' presence and activities are higher with a use of man-made structures, including their homes, workplaces, and transportation

**Data availability statement:** All relevant data are within the paper or in the supplementary file.

**Funding:** This work was supported by the Korea Environment Industry & Technology Institute (KEITI) through the Project for the Development of Biological Diversity Threats Outbreak Management Technology (2018002270004), funded by the Korea Ministry of Environment (MOE). The funders had no role in study design, data collection and analysis, decision to publish, or preparation of the manuscript.

**Competing interests:** The authors declare that the research was conducted in the absence of any commercial or financial relationships that could be construed as a potential conflict of interest.

[1,2]. Thus, they involve the conversion of natural and semi-natural habitats into urban infrastructure and human settlements. This process may influence the population status of many plant and animal species mainly due to habitat loss, habitat fragmentation/isolation, and pollution and degradation of habitat quality [3–6]. Furthermore, urban environments are often characterized by high levels of anthropogenic noise, which can mask or interfere with animal communication signals, especially for animals that use sound to communicate, including amphibians, making it challenging for individuals to detect acoustic cues, and signals may be so garbled that they are unrecognizable or cannot contain any biologically significant information [7–11]. Although individuals of some species developed strategies to mitigate the negative impacts of urban infrastructures and use them in enhancing call emission and propagation, e.g., individuals of the Mientien tree frog (*Kurixalus idiootocus* (Kuramoto & Wang, 1987)) calls emit with longer duration, greater intensity in urban infrastructures [12], most of other amphibians do not show strong correlations between acoustic structure and the surrounding environment [13]. Thus, urban environments may cause a call emitter to lose fitness by failing to draw the attention of a mate or protect its territory and it brings a negative effect of reproduction and survival.

However, vocalization costs energy and may increase predation risk [14–16]. Hence, the calling activity of amphibians is periodic and can be affected by abiotic variables such as temperature and rainfall [17–19]. The abiotic environment is known to shift over time (daily or seasonally) [20,21], yet human factors, including urbanization, can cause abrupt changes in the surrounding environment [22]. Among many other changes, the urban environment may cause high temperatures through heat islands [23]. It is already evident that individuals of many anuran species respond to rapid environmental changes, especially to temperature fluctuations [17,24–28]. For example, individuals of *Allopaa hazarensis* (Dubois & Khan, 1979) and *Nanorana vicina* (Stoliczka, 1872) were found to show quicker metamorphosis, smaller body size, increased frequency of developmental defects or edema, tail kinks, decreased fitness, and increased mortality at higher temperatures [29]. Moreover, temperature is reported to affect the number of males calling seasonally [15,30].

Furthermore, surrounding noise in urban areas may induce higher energy costs and disrupt acoustic communications [30]. It is also reported that calling activities are generally performed in low-noise environments, whereas in response to noise, calls might be stopped or reduced in number [31–33]. In addition to the abiotic factors, biotic factors, such as the presence of predators and population density, may also influence frog calls [15,34]. Thus, amphibians need to appropriately adjust the active time when they can call with less energy by minimizing influences of abiotic and biotic factors as the consequences of human interferences, like urbanization [31–33,35].

Urbanization may also lead to the introduction of invasive species [36–38]. Together with emergence of new habitats as a consequence of urbanization, enhanced human movement and transportation have triggered the recent increase in the spread of invasive species [39]. One of the best examples of such spread of alien invasive species is the American bullfrog (*Aquarana catesbeiana* (Shaw, 1802)), which was introduced into more than 40 nations on four continents and is now

recognized as one of the most impactful invasive amphibian species in the world [40–42]. Like other invasive species, *A. catesbeiana* invasions might also be facilitated by their environmental adaptability [42,43], which indicates that urban areas with changed environments may act as hotspots for their entry and settlement. However, the process by which they interact and cope with changes in urbanized areas, such as higher temperature and noise intensity, remained understudied [43]. Acoustic signaling is highly sensitive to various types of disturbances, making it potentially very important for studying animal behavior, can be examined in a wide variety of ecosystems, recorded, analyzed, synthesized, and played back with reasonable ease and at a low cost [44]. Furthermore, it is already evident that the calling behavior of anurans, unique and exclusive breeding behavior, can be influenced by habitat alteration and anthropogenic noise [45]. Considering the potentiality of using calls in studying anuran reproductive behavior, it may also help understand the impact of urbanization, a new ecosystem with shrunken breeding sites and changed environmental factors, on their breeding phenology [8].

Thus, we studied the vocalization patterns of *A. catesbeiana* individuals, and the number of calls over time, to understand their reproductive behavioral adaptations and breeding strategies in response to urbanization in South Korea to fill gaps in knowledge. We hypothesized that reduced breeding sites, anthropogenic noise, and changed environmental factors in urbanized areas may shape breeding behavior of *A. catesbeiana* through increased competition, interruption by non-natural noise, raised temperature, and alteration in other factors. To test our hypothesis, we calculated the Urbanization Index of 12 potential breeding sites of *A. catesbeiana* and recorded their calls using an Automatic Recording System to observe the influence of urbanization intensity on breeding behavior.

## 2. Materials and methods

### 2.1. Studied species

The American bullfrog, *Aquarana catesbeiana,* previously known as *Rana* (*Lithobates*) *catesbeiana*, belongs to the order Anura and family Ranidae (Fig 1a). They are considered one of the 100 world's worst invasive alien species and has dispersed over 40 countries (Fig 1b) [40]. Studies have reported an association between the American bullfrog and increased *Batrachochytrium dendrobatidis* (Bd) prevalence and population declines in many amphibian species [46]. Furthermore, a recent study showed that at least 84% of native anurans in the South Korea are threatened by *A. catesbeiana* [42]. Similar to other invaded countries, this species was brought from the US and Japan to the South Korea during the 1950s and 1970s, mostly to serve as a food source [47]. Now, *A. catesbeiana* has established local populations and can be observed all over the country and can be easily found in rivers and reservoirs [48,49]. The breeding season of the species may last from June to August in the South Korea [19,42,48]. Male bullfrogs exhibit courtship behaviors, especially through calls, during the breeding season. They typically call for 0.6 to 1.5 seconds near the edge of their habitat, either alone or in choruses with several males [14,50–53].

We did not catch or kill any individual but only recorded their calls in different areas. Given the study focused on an 'ecosystem disturbance' species [42] and in a non-lethal way, it did not require any permission.

### 2.2. Sampled sites

After confirming the breeding activities of bullfrogs, we selected 12 study sites in Gwangju and Naju, in South Jeolla Province, South Korea (Figs 1c and 1d). The study sites were categorized according to the degree of urbanization, using Arcmap 10.8.1 (Esri, USA) and the Environmental Geographic Information Service (EGIS, https://egis.me.go.kr). The degree of urbanization was calculated using the percent of buildings in the land cover of a 1.6 km radius of each site [54,55]. The area of 1.6 km radius was selected to cover the total home range of bullfrogs [42,54]. We followed de Satgé et al. (2019) method with a slight modification to apply it in Korea for measuring grades of urbanization and categorized the sites into three types: type 1, 0–3%; type 2, 5–10%; and type 3, 13% or above [55]. Four sites were selected for each type (Table 1). Type 1 included mostly rice paddies, fields, and mountains, whereas type 3 included mostly buildings (Fig 1d).

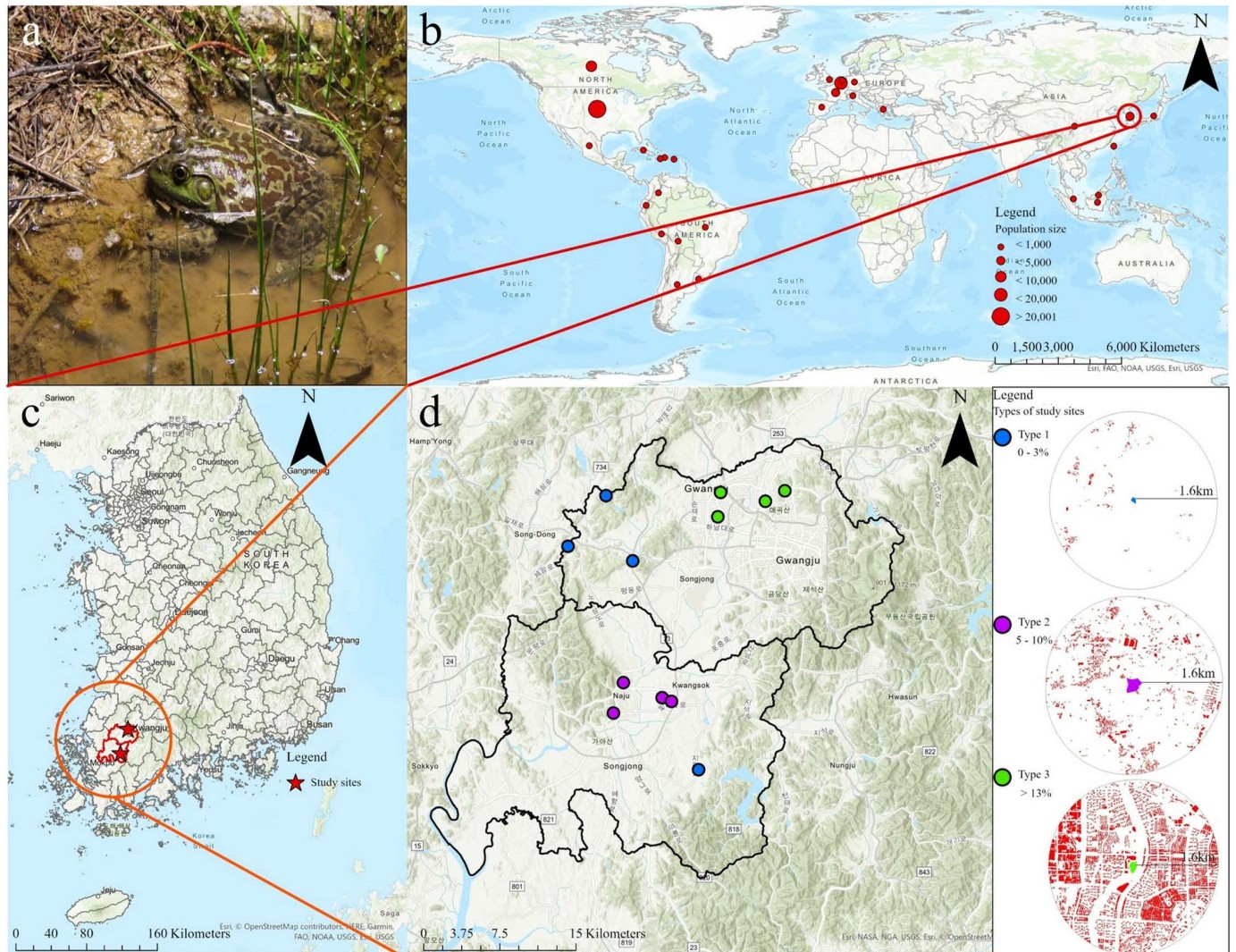

**Fig 1. Study species and sites. (a)** Invasive American Bullfrog (*Aquarana catesbeiana*). **(b)** The worldwide distribution of *A. catesbeiana* populations. The data were retrieved from GBIF (https://www.gbif.org/). **(c)** Map of the South Korea showing the study areas. **(d)** Studied reservoirs in Gwangju and Naju-si with urbanization index represented by color (see side panel). The maps were generated using Arcmap 10.8.1 (ESRI, USA; https://support.esri. com/en/products/desktop/arcgis-desktop/arcmap/10-8-1).

## 2.3. Data collection

We used Song Meter 4 (SM4, Wildlife Acoustic, USA) at a sample rate of 12 kHz and a resolution of 16 bits per sample. Recording was conducted for 162 days from April 26 to October 5, 2023, matching with the breeding period of the bullfrog in the South Korea [19,42]. Initially, we surveyed and selected the reservoirs based on the previous reports on the presence of bullfrogs [49]. The recorders were installed after the confirmation of the presence of the bullfrogs at the study sites during the present surveys. Thus, there was a difference in the recording period from site to site. The earliest installation of the recorders was on 26 April 2021, whereas the latest installation of the recorders was on 15 May 2021 (S1 Table). The SM4 were installed about 1 m above ground level on the nearest tree from the reservoir edge. The distance of the tree ranged from 0 to 1m. Considering the longer study period, we recorded first 5 minutes every hour for 24 hours

**Table 1. Study sites at Gwangju and Naju in the South Korea, with their locations and degrees of urbanization.**

| Group | Site | Degree of urbanization (%) | Latitude (N) | Longitude (E) |
|---|---|---|---|---|
| **Type 1** | Dongnim | 0.76 | 35.2068 | 126.6995 |
| | Unpyeong | 0.88 | 35.1491 | 126.7290 |
| | Bokman | 1.70 | 35.1615 | 126.6590 |
| | Songhyeon | 2.07 | 34.9642 | 126.8020 |
| **Type 2** | Ogang | 5.83 | 35.0279 | 126.7620 |
| | Naedong | 6.47 | 35.0141 | 126.7090 |
| | Useok | 7.29 | 35.0247 | 126.7720 |
| | Daeho | 9.60 | 35.0410 | 126.7200 |
| **Type 3** | Hansaebong | 13.24 | 35.2123 | 126.8933 |
| | Yangsan | 19.47 | 35.2028 | 126.8721 |
| | Dochon | 21.23 | 35.2104 | 126.8236 |
| | Suwan | 23.80 | 35.1887 | 126.8209 |

throughout the breeding season to make the study cost effective and unified. The SD cards and batteries were collected and replaced once a month. We used data from May to September for seasonal patterns and October to confirm the end of calling activity.

A total of 14 environmental, time, biotic, anthropogenic, and site variables were selected to identify the factors affecting the calling activity of bullfrogs [24,26,56]. Environmental variables included air temperature (AT), water temperature (WT), humidity (H), rainfall (R), and wind speed (WS). Temperature and humidity were recorded through a temperature and humidity data logger (RC-4HC, Elitech Technology, USA) installed at the same location as the SM4. Another data logger was installed at the water's edge at each site and measured the water temperature at a 15 cm depth. The data loggers were recorded every 30 minutes. Rainfall and wind speeds were based on data from the Korean Meteorological Administration (KMA, https://www.weather.go.kr), which was based on stations close to each site. Rainfall and wind speeds were recorded every hour. These recorded temperatures at a resolution of 0.1°C, humidity to 0.1 percentage point, rainfall to 0.1 mm, and wind speed to 0.1 m/s (S2 Table and S1 Fig).

Time variables included the Julian date (JD), time (T), and sunrise–sunset time (ST). The Julian date was recorded from 115 to 278 and time was recorded from 0 to 23. The Julian date was used to analyze the seasonal variation, and time for the daily variation in bullfrog calling activity. The sunrise–sunset time from the Korea Astronomy and Space Science Institute (KASI, https://astro.kasi.re.kr) was used to distinguish between day and night: 0 = from sunset to sunrise, night; 1 = from sunrise to sunset, day. As biotic variables, we included the most common species in Korea, the black-spotted pond frog (*Pelophylax nigromaculatus* (Hallowell, 1861), PC) and Japanese tree frog (*Dryophytes japonicus* (Günther, 1859), DC). The calls of these individuals were measured with the Calling Index (CI), as described in the next subsection. The anthropogenic variables were traffic noise and human-presence noise. We counted the number of traffics to check the level of noise from passing vehicles and recorded the presence and absence of humans through the noise made by passersby. The level of noise was increased with traffic volume and human presence [57]. The level of noise was measured by counting the noise events. All the recorded sounds (through SM4), a total of 25,544 minutes in 162 days, we analyzed the 2nd – 3rd minute of recorded time to measure the noise in the study areas. The traffic noise was considered as "1" when car passed by [57,58], while the intensity of noise was calculated as decibels below full scale (dBFS) using the Raven pro 1.6 (Cornell Lab of Ornithology, NY) between 90 Hz and 7000 Hz [59]. As zero dBFS represents maximum power, values for inband power are negative, with negative numbers closer to zero indicating more power [60]. We considered continuous noise (CN) if the aircraft and light vehicles passed by with high speed with a noise intensity of −46.79 ~ −3.23 dBFS, 90 ~ 5,668 Hz, for more than 20 seconds. In such cases, we considered it as maximum "3". On the other hand, we

considered discontinuous noise (DN) when vehicles passed by in a constant low speed, irrespective of types [61], with a noise intensity of −46.92~−11.2 dBFS, 90~6124.3 Hz, for less than 20 seconds. Human noise (HN) was considered as "1" when human made sound (e.g., voice and footsteps) were heard within a 1-minute period and "0" when they were not. To avoid biases, a single person (the 1st author) heard and measured the noise level from the whole recordings. Traffic noise and human noise were counted using the recording file with Raven pro. Site (SI) variables were study sites, and these are the grades of urbanization from 1 to 12.

## 2.4. Call analysis

We used Raven Pro 1.6 to visualize calls and count them. To avoid biases and get the best quality data, the number of individuals was estimated by bullfrog advertisement calls over 2:00–3:00-minute periods in the recording file. The number of individuals was counted by measuring the Calling Index (CI). The CI was measured with values from 0 to 4: 0 = no individual called for 1 minute, 1 = one individual called for 1 minute, 2 = two individuals called for 1 minute, 3 = three individuals called for 1 minute, and 4 = four or more individuals' or indistinguishable calls for 1 minute [26]. Since anurans have different call characteristics that can be used in species [62], population [9] and individual identifications [63], each individual was identified through sonogram analysis. After checking all recorded files, we analyzed a total of 7,436 minutes, excluding files where the sonogram could not be identified due to strong noise such as rain, or that did not contain any bullfrog calls. We identified 3,341 minutes in type 1, 2,186 minutes in type 2, and 1,909 minutes in type 3.

## 2.5. Statistical analyses

After confirming the von Mises distribution of our dataset ($p < 0.05$), the seasonal and daily calling activity of bullfrogs were analyzed through circular statistical analysis using Oriana 4 (Kovach Computing Services, Wales, UK). We calculated the mean vector (μ), mean vector length (r), median, concentration, circular variance, and circular standard deviation. To look for periodicity and irregularity in bullfrog call data, we used the Rayleigh test for uniformity. Furthermore, we conducted Watson–Williams $F$ tests to compare the means, and Mardia tests to compare the significant distribution difference.

 We used Generalized Linear Mixed Effects Models (GLMMs) to see the relationship between call activity and surrounding environmental, biotic, and abiotic factors and identify variables influencing the calling patterns of bullfrogs at the different study sites. We conducted the GLMMs using a Multinomial Logistic Regression because the CI of the bullfrog, which is a dependent variable, is a discrete variable ranging from 0 to 4. Except for site, which we defined as a random effect, we defined all other variables as fixed effects. Since there was no significant correlation (r > 0.8) and no multicollinearity between the 14 variables, all variables were included in the analysis (S3 Table). We created 19 models with a combination of 14 variables (S4 Table). We used Akaike's Information Criterion (AIC) to select the optimal model (S4 Table). We analyzed the AIC and performed the GLMMs using SPSS Statistics 23 (IBM Corp. Armonk, NY, USA).

## 3. Results

### 3.1. Seasonal pattern of calling activity

Our results revealed a significant seasonality throughout the entire recording period in all study sites (Rayleigh test $p < 0.05$; Table 2). We found an early onset of calling activities in urbanized *A. catesbeiana* populations, which suggests an urban-induced shift in breeding phenology. The highest CI output occurred in July at type 1 (lower urban sites) and in June at type 2 (medium urban sites) and type 3 (higher urban sites; Fig 2a). There were also differences in average active days between type 1 and type 3 (Watson–Williams test, $F = 16,476$, $p < 0.001$), and type 2 and type 3 (Watson–Williams test, $F = 6,316$, $p = 0.013$) sites. The number of active days was longest at type 1 sites and shortest for type 3. Additionally, we also found a difference in seasonal calling activity patterns between type 1 and type 3 (Mardia test, $W = 11.547$, $p = 0.003$).

**Table 2. Circular statistical analysis of *Aquarana catesbeiana* calling activity in different types of study sites.**

| Variables | Type 1 | Type 2 | Type 3 |
|---|---|---|---|
| *Seasonal patterns* | | | |
| No. of observations | 124 | 108 | 87 |
| Mean vector (μ) | 190.58° | 166.532° | 159.159° |
| Mean date (yy/mm/dd) | 21/07/13 | 21/06/18 | 21/06/11 |
| Length of mean vector (r) | 0.869 | 0.931 | 0.965 |
| Concentration | 4.111 | 7.491 | 14.699 |
| Circular variance (S) | 0.131 | 0.069 | 0.035 |
| Circular standard deviation (SD) | 30.411° | 21.718° | 15.213° |
| Rayleigh test (Z) | 75.449 | 44.175 | 48.46 |
| Rayleigh test (p) | < 0.0001 | < 0.0001 | < 0.0001 |
| *Daily patterns* | | | |
| No. of observations | 3523 | 2324 | 1927 |
| Mean vector (μ) | 22.042° | 26.74° | 24.961° |
| Mean time (hh:mm) | 01:28 | 01:46 | 01:39 |
| Length of mean vector (r) | 0.709 | 0.679 | 0.646 |
| Median | 30° | 30° | 30° |
| Concentration | 2.064 | 1.883 | 1.713 |
| Circular variance (S) | 0.291 | 0.321 | 0.354 |
| Circular standard deviation (SD) | 47.507° | 50.43° | 53.562° |
| Rayleigh test (Z) | 4779.385 | 2293.623 | 1948.042 |
| Rayleigh test (p) | < 0.0001 | < 0.0001 | < 0.0001 |

There is a difference in the seasonal distribution between type 1 and type 2, and type 1 and type 3. Type 1 had a decreasing pattern after peak in July, and type 3 had a decreasing pattern after peak in June.

### 3.2. Daily pattern of calling activity

Bullfrogs' calls were recorded all day and night, however, most calling activity was observed at night (20:00–06:00). The highest CI output was found at 03:00 at type 1 and type 2, and 02:00 at type 3 sites (Rayleigh test $p < 0.05$; Table 2; Fig 2b, c and d). The average activity times showed differences between type 1 and type 2 sites (Watson–Williams test $F = 7.61$, $p = 0.006$). The average activity time was at 01:28 in type 1, 01:46 in type 2 and 01:39 in type 3. The daily pattern of calling activity was significantly different between type 1 and type 3 (Mardia test $W = 23.126$, $p < 0.001$) and between type 2 and type 3 (Mardia test $W = 78.448$, $p < 0.001$) sites. There is a difference in the 24-hour activity distribution between type 1 and type 2, and type 1 and type 3. Calling activity peaked at around dawn in all types but was more widely distributed during the daytime at type 3 sites.

### 3.3. Factors influencing calling activity

We checked the influence of 14 factors on the bullfrog calling index in the study sites. In the top five models, the abiotic model with continuous noise and water temperature had the lowest AIC (Table 3). Results of GLMM showed that there was more calling activity when the noise was less, and the water temperature was lower ($p < 0.001$; Table 4; Fig 3a and b). We found a decreased calling activity when the water temperature was more than 27°C and less than 16°C. We recorded as few as 34 calls during continuous traffic noise (CN = 3), whereas, during no continuous traffic noise (CN = 0), the number of calls was as high as 7,430. This means the more traffic the less call activity. Also, our results suggest a negative correlation between CI and noise intensities (Fig 3c).

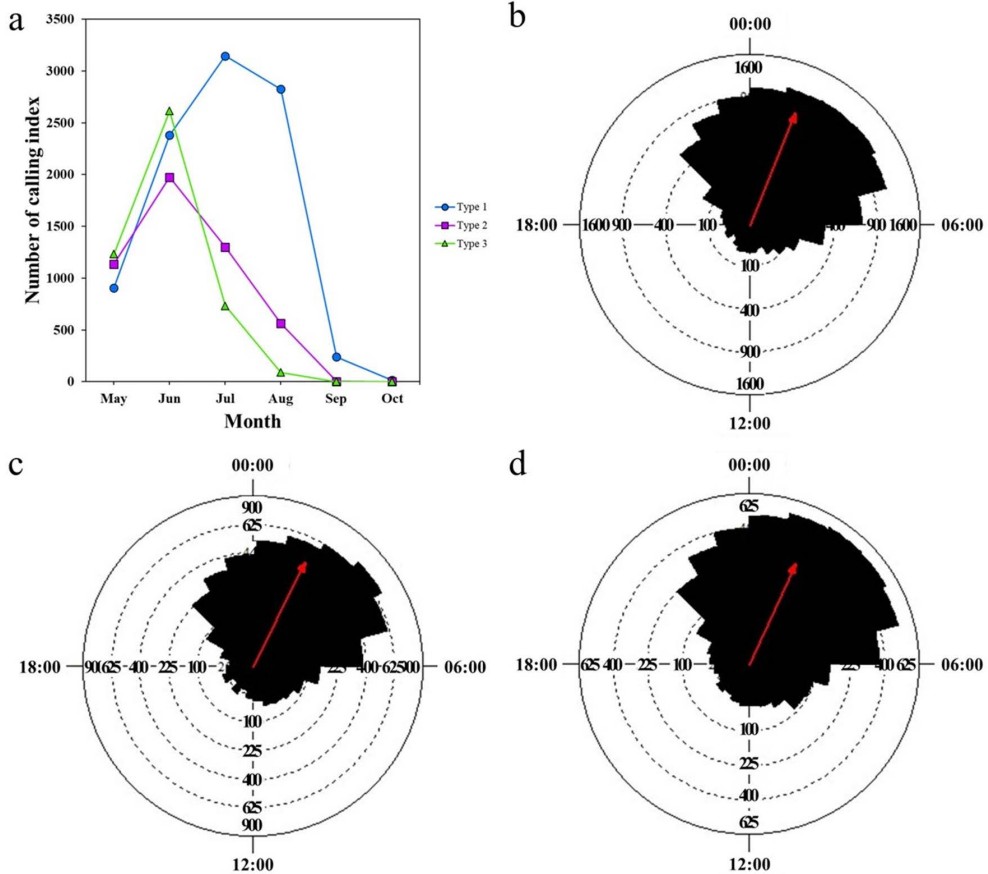

**Fig 2. Seasonal and daily calling activity patterns of the bullfrog in different types of study sites. (a)** The seasonal calling pattern. Circles represent habitat type 1 (blue), squares habitat type 2 (magenta), and triangles habitat type 3 (green). **(b)** The daily calling pattern in type 1 sites. **(c)** The daily calling pattern in type 2 sites. **(d)** The daily calling pattern in type 3 sites. The length of the sections indicates the calling index of bullfrogs in the corresponding time range. The red line depicts the mean vector length **(r)**, which indicates the concentration of species' calling activity during the day.

**Table 3. Comparison of the top five alternative models ranked by AIC to explain the influence of factors on bullfrog calling activity.**

| No | Model | Variable | k[1] | AIC[2] | delta AIC[3] | w[4] |
|---|---|---|---|---|---|---|
| 1 | Abiotic | CN, WT | 4 | 3,975.92 | 0 | 1 |
| 2 | Abiotic, Date, and Time | WT, CN, ST | 5 | 5,583.93 | 1,608.02 | 0 |
| 3 | Biotic, Environmental | PC, WT | 4 | 6,127.02 | 2,151.10 | 0 |
| 4 | Biotic, Date, and Time | PC, T | 4 | 8,513.53 | 4,537.61 | 0 |
| 5 | Anthropogenic, Date, and Time | CN, T | 4 | 8,688.87 | 4,712.95 | 0 |

1 Number of estimate parameter.

2 Akaike's information criterion (AIC).

3 Difference between the top model and given model.

4 Akaike weight.

CN, continuous traffic noise; PC, *Pelophylax nigromaculatus* calling index; ST, sunset and sunrise times; T, time; WT, water temperature.

**Table 4. Parameter estimates (coefficients and standard error) from the best-supported model for bullfrog calling activity.**

| Variable | Coefficient | SE | t | p |
|---|---|---|---|---|
| *Bullfrog calling index 1* | | | | |
| (intercept) | −1.132 | 0.198 | −5.716 | < 0.001* |
| CN | −0.174 | 0.053 | −3.264 | 0.001* |
| WT | −0.032 | 0.008 | −4.084 | <0.001* |
| *Bullfrog calling index 2* | | | | |
| (intercept) | −0.557 | 0.232 | −2.399 | 0.016* |
| CN | −0.450 | 0.087 | −5.161 | < 0.001* |
| WT | −0.072 | 0.009 | −7.564 | < 0.001* |
| *Bullfrog calling index 3* | | | | |
| (intercept) | −0.211 | 0.257 | −0.822 | 0.411 |
| CN | −0.996 | 0.171 | −5.843 | < 0.001* |
| WT | −0.096 | 0.011 | −9.009 | < 0.001* |
| *Bullfrog calling index 4* | | | | |
| (intercept) | 1.303 | 0.177 | 7.382 | < 0.001* |
| CN | −2.019 | 0.252 | −7.998 | < 0.001* |
| WT | −0.123 | 0.007 | −16.762 | < 0.001* |

*= significant.

CN, continuous traffic noise; WT, water temperature.

## 4. Discussion

The calling activity patterns of the American bullfrog, a prolonged breeder, in natural and urban environments were observed in the current study. Bullfrogs have a long breeding season, from April to August [19,42]. However, we found them most active in June and July. The calling activity patterns were different among the three types of study areas categorized based on the grades of the urbanization.

### 4.1. Seasonal pattern of calling activity

We found sharp differences in seasonal activity, the onset of breeding season, and the daily calling behavior in *A. catesbeiana* between highly urbanized and non-urbanized areas. Although studies have reported June to August as the breeding season and calling activity period for bullfrogs in South Korea [19,42,48], we recorded calls in late April to early October. However, we observed the onset of the calls in mid-May in most of the urbanized study sites (S1 Table). This might have resulted from the longer study period and continuous call recording used in our study and suggests a much longer breeding season for bullfrogs than the current knowledge posits. It also indicates that the areas with lower urbanization levels have a longer breeding period, running later into the season, and a later peak time. These differences in seasonal activity patterns, longer or shorter, might be attributed to the variations in the surrounding environmental factors of each habitat type [15]. For instance, more intense selection pressure and evolutionary conditions in urban environments due to artificial infrastructures, pollution, changes in abiotic factors, etc. may lead to behavioral modifications in urban dwelling species [64]. Changes in breeding phenology and duration may be such behavioral modifications, allowing bullfrogs in the urban environment to start breeding early and finish it quickly [5,64,65]. Previous studies have also found the breeding phenology of many amphibian species to be strongly dependent on the surrounding conditions [66–69].

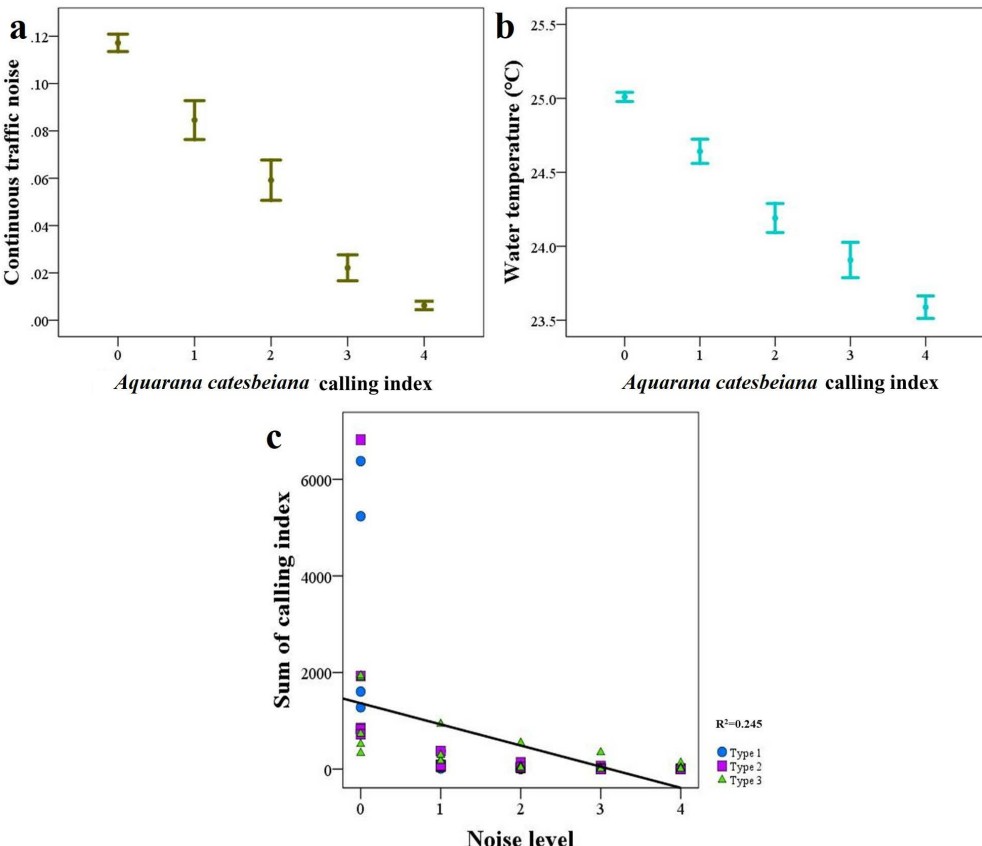

**Fig 3. Important factors affecting the bullfrog calling index in study sites. (a)** The influence of continuous noise, which was included in the top-ranked GLMM predicting the calling index. **(b)** The influence of water temperature, which was also included in the top-ranked GLMMs. **(c)** The relationship between calling index and noise intensities.

## 4.2. Daily patterns of calling activity

In this study, calling activity of *A. catesbeiana* was mostly observed between 20:00 and 6:00. This finding is similar to those of previous reports, where researchers found nocturnal calling activities of *A. catesbeiana*, with the highest concentration of calls recorded at dawn [70]. Additionally, many other anurans, including other species of the genus *Aquarana*, also show nocturnal calling with the most activity at dawn [71,72]. Generally, this might be a strategy to reduce the risk of dryness and dehydration [73,74]. It might also be attributed to the lower temperature and less ambient noise during the night than in the daytime, which is conducive to efficient sound transmission [75].

However, the different types of study areas did not show significant differences in average daily activity periods but did in activity duration. Types 1 and 2 showed a similar amount of daily activity time, but type 3 showed a longer calling activity period. Additionally, some parts of the type 3 sites showed fierce population-level competition due to the narrowness of the reservoirs and the high numbers of individuals. Thus, there was a higher intensity of diurnal calls. It might be the strategy to avoid competition for breeding partners. Many authors have reported the formation of individuals forming a certain sphere of influence on the edge of the pond through breeding calls [51], a phenomenon of competition, and individuals attempting to avoid or reduce this competition by choosing temporal differences, even diurnal and nocturnal breeding activities [45,76].

## 4.3. Factors influencing bullfrog calling activity

Continuous traffic noise and water temperature were the most influential factors affecting the bullfrog's calling activity in our study. We observed decreased calling activities on nights with more noise, especially, we found the lowest CIs in type 3, which had high levels of continuous noise. This might be an indication of a coping mechanism to deal with anthropogenic and traffic noises in urban areas. Given that auditory masking by anthropogenic noise can reduce an individual's ability to detect voice, interfering with acoustic communications [77], the number of CIs and the participation of males in choruses can be negatively impacted [7,78]. Reports on the impacts of urban noise on animals' acoustic communication [78–80], including the impact of traffic noise on the calling activities of anurans [10,33], support our findings. In addition, the frequency ranges of traffic sounds and bullfrog calls also explain our results. The bullfrog's call has a frequency between 90 and 4000 Hz, which falls entirely within the frequency range of traffic noise, 50–7000 Hz; thus, making it easily interrupted [22,81]. It might also be a strategy to avoid excessive energy consumption by reducing calling activity during continuous traffic noise [82]. Being a cosmopolitan invader, A. catesbeiana seems to show high plasticity to cope with urban environments, arranging calling activities according to the intensity of ambient noise [82,83].

The influence of temperature on anuran physiology and behavior—such as calling activity, especially in temperate monsoon climates—is also not surprising [24,84,85]. Previous observations on bullfrogs support our findings on the influence of temperature on their calling activity [86]. Our study suggests the water temperature ranges for A. catesbeiana calling activity ranges from 16°C to 30°C. Although studies indicate that bullfrogs are a summer (warmer weather) breeder, with water temperature (which is also related to the air temperature) portrayed as a potential predictor of calling activity, the lower threshold for water temperature was reported to vary [24]. However, many previous studies have found results similar to ours [24,87]. The differences in the findings might be attributed to the environmental conditions of the areas used in different studies. Thus, although a few studies have suggested a minor impact of short-term weather, such as temperature, on prolonged breeders [26,88], its influence is well-evident in bullfrogs' calling patterns than in many others [89,90].

Together with the factors like humidity, rainfall, wind speed, presence or absence of the predators etc., much research has reported on the relationship between temperature and the onset of breeding phenology in amphibians [91,92]. Unlike other species, though A. catesbeiana depends on the seasonal temperature, it begins calling when the water temperature raises with the surrounding air temperature and reaches a suitable range [92]. During this study, type 3 sites (the most urbanized) showed a higher air and water temperature than other types (S1 Fig), and thus, this may have influenced early breeding phenology and calling activities. Notably, previous studies have also found that many amphibian species start breeding earlier in urban areas than in rural areas, sometimes with longer breeding periods in rural areas [8,93,94]. Furthermore, urbanization causes variations in environmental factors, such as increasing air temperatures through heat islands, that are crucial for amphibian calling activity and breeding phenology [15,23,30,95].

Thus, our study concluded that the bullfrog's calling activity is highly influenced by continuous traffic noise and water temperature caused by urban environment. It also supports the existence of urbanization-induced patterns in the calling activities and breeding periods of amphibians. Although this study could not observe the subsequent impacts of shifts in breeding phenology, it sheds light on the effect of urbanization on breeding activities and how an invasive species copes with the changed environment in new areas. We recommend further intensive long-term research to better determine the onset of the breeding phenology and the extent to which the breeding period changes in different levels of urbanized and natural habitats and further explore the related real consequences. However, this study will help future researchers in studying the impacts of urbanization on biodiversity and animal behavior. Additionally, our research provides information on the most active times for A. catesbeiana in South Korea and when they are easily traceable (i.e., through calls) in all habitat types, which is important for identifying their population status and setting effective eradication measures in its invasive ranges [96–98].

## Supporting information

**S1 Fig. Environmental variables at the study sites, categorized by their levels (Type).** (A) air temperature, (B) humidity, (C) water temperature, (D) rainfall, and (E) wind speed. The data represents the daily average in each recording period.
(PDF)

**S1 Table. Recording periods in each site.**
(PDF)

**S2 Table. Environmental variables in the study sites categorized by different urbanization levels (site type).** The data are means±SD (min–max). Temperature is in °C, humidity in %, rainfall in mm, and wind speed in m/s.
(PDF)

**S3 Table. The relationships between the 14 environmental variables and level of urbanization.** The numbers shown are Spearman correlation coefficients.
(PDF)

**S4 Table. The full model combining all 14 variables and alternative models containing various combinations of the 14 variables.** All models are ranked by AIC for their ability to explain the influence of the factors on bullfrog calling activity.
(PDF)

## Acknowledgments

The authors would like to thank everyone involved in the data collection and analysis.

## Author contributions

**Conceptualization:** Ji-A Lee, Md Mizanur Rahman, Ha-Cheol Sung.

**Data curation:** Ji-A Lee, Seung-Ju Cheon.

**Formal analysis:** Ji-A Lee.

**Funding acquisition:** Ha-Cheol Sung.

**Investigation:** Ji-A Lee, Seung-Ju Cheon.

**Methodology:** Ji-A Lee.

**Software:** Ji-A Lee.

**Supervision:** Md Mizanur Rahman, Ha-Cheol Sung.

**Validation:** Md Mizanur Rahman, Amaël Borzée.

**Visualization:** Ji-A Lee, Md Mizanur Rahman.

**Writing – original draft:** Ji-A Lee, Md Mizanur Rahman.

**Writing – review & editing:** Md Mizanur Rahman, Seung-Ju Cheon, Amaël Borzée, Ha-Cheol Sung.

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
