## [Decision Letter · Decision Letter 0]

Apr 28 2025

Dear Dr. Sung,

Thank you for submitting your manuscript to PLOS ONE. After careful consideration, we feel that it has merit but does not fully meet PLOS ONE’s publication criteria as it currently stands. Therefore, we invite you to submit a revised version of the manuscript that addresses the points raised during the review process.

Please submit your revised manuscript by Apr 28 2025 11:59PM If you will need more time than this to complete your revisions, please reply to this message or contact the journal office at plosone@plos.org . A rebuttal letter that responds to each point raised by the academic editor and reviewer(s). You should upload this letter as a separate file labeled 'Response to Reviewers'.A marked-up copy of your manuscript that highlights changes made to the original version. You should upload this as a separate file labeled 'Revised Manuscript with Track Changes'.An unmarked version of your revised paper without tracked changes. You should upload this as a separate file labeled 'Manuscript'.

We look forward to receiving your revised manuscript.

Kind regards,

Daniel de Paiva Silva, Ph.D.

Academic Editor

PLOS ONE

Journal Requirements:

“This work was supported by the Korea Environment Industry & Technology Institute (KEITI) through the Project for the Development of Biological Diversity Threats Outbreak Management Technology (2018002270004), funded by the Korea Ministry of Environment (MOE).”

4. Please note that funding information should not appear in the Acknowledgments section or other areas of your manuscript. We will only publish funding information present in the Funding Statement section of the online submission form. Please remove any funding-related text from the manuscript. 

5. We note that Figure 1 in your submission contain map/satellite images which may be copyrighted. All PLOS content is published under the Creative Commons Attribution License (CC BY 4.0), which means that the manuscript, images, and Supporting Information files will be freely available online, and any third party is permitted to access, download, copy, distribute, and use these materials in any way, even commercially, with proper attribution. For these reasons, we cannot publish previously copyrighted maps or satellite images created using proprietary data, such as Google software (Google Maps, Street View, and Earth). For more information, see our copyright guidelines: http://journals.plos.org/plosone/s/licenses-and-copyright.

1) You may seek permission from the original copyright holder of Figure 1 to publish the content specifically under the CC BY 4.0 license.  

2) If you are unable to obtain permission from the original copyright holder to publish these figures under the CC BY 4.0 license or if the copyright holder’s requirements are incompatible with the CC BY 4.0 license, please either i) remove the figure or ii) supply a replacement figure that complies with the CC BY 4.0 license. Please check copyright information on all replacement figures and update the figure caption with source information. If applicable, please specify in the figure caption text when a figure is similar but not identical to the original image and is therefore for illustrative purposes only.

**Additional Editor Comments:**

Dear Dr. Sung,

After this first review round, both reviewers reached very positive views regarding your study. Therefore, I am recommending a minor review before the second (and possibly last) review round. Please consider all suggestions made by the reviewers to improve the next version of your manuscript.

Sincerely,

Daniel Silva

Reviewers' comments:

Reviewer's Responses to Questions

**Comments to the Author**

1. Is the manuscript technically sound, and do the data support the conclusions?

Reviewer #1: Yes

Reviewer #2: Yes

2. Has the statistical analysis been performed appropriately and rigorously?

Reviewer #1: Yes

Reviewer #2: Yes

3. Have the authors made all data underlying the findings in their manuscript fully available?

Reviewer #1: Yes

Reviewer #2: Yes

4. Is the manuscript presented in an intelligible fashion and written in standard English?

Reviewer #1: Yes

Reviewer #2: Yes

Reviewer #1: The aim of the study was to evaluate the impact of the urbanization on the calling activity pattern of an invasive frog species. Overall, the study is well designed and provides data to respond to the proposed objectives. The introduction, methods, results and discussion are well written and have a good theoretical basis, although I think the number of citations could be greatly reduced (more than 100 references used). Interesting results were found, such as a reduction in the time of calling activity and an earlier onset of the breeding period in urban bullfrog populations. In addition to carrying pathogens, bullfrogs may also represent acoustic overlap (niche) with native anuran species, as demonstrated by Medeiros et al. (2017 - doi: 10.1007/s10530-016-1327-7). All these findings may help to manage and control the populations of this invasive species. Therefore, this study is interesting for researchers who study the impacts of urbanization on biodiversity, for the ecology of invasive species and for herpetologists.

Still, some points require special attention. For example, regarding the use of the calling indices of P. nigromaculatus and D. japonicus. Nothing in the Introduction or in the study objectives provides support for evaluating the influence of the presence of other anuran species on the calling and reproduction activity of the American bullfrog.

In the attached file I have included some important suggestions and comments to improve the manuscript. Therefore, the study needs some minor revisions before being accepted for publication.

Reviewer #2: The proposed article addresses a topic of great ecological interest, as it deals with the study of an invasive species of extreme importance. Furthermore, through the study of these populations, the authors shed light on the problem that the process of urbanization poses to these animal populations, and their results can be used as a tool to help understand and also to propose mitigating measures when this issue is involved. The study as a whole was very well conducted, and the methods used to obtain the results were also very well applied. I would like to make a small remark regarding the discussions related to your main finding, which is the influence of noise on the behavior of the studied anurans (for more details, see the body of the text), as well as other corrections/suggestions made throughout the text. I am available for any clarifications.

**Do you want your identity to be public for this peer review?** For information about this choice, including consent withdrawal, please see our Privacy Policy

Reviewer #1: No

Reviewer #2: No

---

## [Author Response · Author response to Decision Letter 1]

27 Apr 2025

Point-by-point Response to the Editor’s and Reviewers’ comments

Editor’s Comments:

Comment:

After this first review round, both reviewers reached very positive views regarding your study. Therefore, I am recommending a minor review before the second (and possibly last) review round. Please consider all suggestions made by the reviewers to improve the next version of your manuscript.

Response:

We highly appreciate your efforts. We are glad that although the review process took a long time, it came up with very positive result. We hope next rounds of the publication of our manuscript will be faster and smoother. We thank you for your affirmative recommendation for our manuscript. We considered all suggestions made by the reviewers to improve the revised version of our manuscript. Herein, we are attaching a point-by-point response to the reviewers’ comments with details of the corrections and modifications we made to our manuscript. Please check and let us know if you have any additional comments. If necessary, we are open to revise our manuscript further to improve it to fulfill the high standard of your journal and editorial process.

PLOS ONE

Journal Requirements:

Comment:

Response:

Thanks for your comment. We have revised our manuscript accordingly.

Comment:

Response:

We added it in lines 129 – 130.

Comment:

“This work was supported by the Korea Environment Industry & Technology Institute (KEITI) through the Project for the Development of Biological Diversity Threats Outbreak Management Technology (2018002270004), funded by the Korea Ministry of Environment (MOE).”

Response:

We highly appreciate your valuable advice. We have stated the role of the funders clearly and mentioned it in the cover letter as per your instructions.

Comment:

4. Please note that funding information should not appear in the Acknowledgments section or other areas of your manuscript. We will only publish funding information present in the Funding Statement section of the online submission form. Please remove any funding-related text from the manuscript.

Response:

Thanks for your comment. We have deleted the funding information from the manuscript.

Comment:

5. We note that Figure 1 in your submission contain map/satellite images which may be copyrighted. All PLOS content is published under the Creative Commons Attribution License (CC BY 4.0), which means that the manuscript, images, and Supporting Information files will be freely available online, and any third party is permitted to access, download, copy, distribute, and use these materials in any way, even commercially, with proper attribution. For these reasons, we cannot publish previously copyrighted maps or satellite images created using proprietary data, such as Google software (Google Maps, Street View, and Earth). For more information, see our copyright guidelines: http://journals.plos.org/plosone/s/licenses-and-copyright.

1) You may seek permission from the original copyright holder of Figure 1 to publish the content specifically under the CC BY 4.0 license.

2) If you are unable to obtain permission from the original copyright holder to publish these figures under the CC BY 4.0 license or if the copyright holder’s requirements are incompatible with the CC BY 4.0 license, please either i) remove the figure or ii) supply a replacement figure that complies with the CC BY 4.0 license. Please check copyright information on all replacement figures and update the figure caption with source information. If applicable, please specify in the figure caption text when a figure is similar but not identical to the original image and is therefore for illustrative purposes only.

Response:

We appreciate your concern. The maps were generated using Arcmap 10.8.1 (ESRI, USA; https://support.esri.com/en/products/desktop/arcgis-desktop/arcmap/10-8-1), which is free to use and now, we referred the source in the figure caption. Please, check lines 108 – 113 of the revised manuscript. We did the same thing in one of our recent papers published in PlosOne, https://doi.org/10.1371/journal.pone.0281808.g001.

Comment:

Response:

Thanks for your suggestion. We have included captions for our Supporting Information files at the end of our manuscript and updated in-text citations following your suggestion.

Comment:

Response:

We appreciate your concern. We have checked and revised the reference list accordingly.

Comments from Reviewer #1:

Comment:

Reviewer #1: The aim of the study was to evaluate the impact of the urbanization on the calling activity pattern of an invasive frog species. Overall, the study is well designed and provides data to respond to the proposed objectives. The introduction, methods, results and discussion are well written and have a good theoretical basis, although I think the number of citations could be greatly reduced (more than 100 references used). Interesting results were found, such as a reduction in the time of calling activity and an earlier onset of the breeding period in urban bullfrog populations. In addition to carrying pathogens, bullfrogs may also represent acoustic overlap (niche) with native anuran species, as demonstrated by Medeiros et al. (2017 - doi: 10.1007/s10530-016-1327-7). All these findings may help to manage and control the populations of this invasive species. Therefore, this study is interesting for researchers who study the impacts of urbanization on biodiversity, for the ecology of invasive species and for herpetologists.

Still, some points require special attention. For example, regarding the use of the calling indices of P. nigromaculatus and D. japonicus. Nothing in the Introduction or in the study objectives provides support for evaluating the influence of the presence of other anuran species on the calling and reproduction activity of the American bullfrog.

In the attached file I have included some important suggestions and comments to improve the manuscript. Therefore, the study needs some minor revisions before being accepted for publication.

Response:

We highly appreciate your insightful comments and outlining the strength and significance of our manuscript. However, we have read your comments thoroughly and revised our manuscript accordingly. In addition, we have reviewed and reduced the number of references following your suggestion. Please check this rebuttal to explore the changes and corrections we made to our manuscript in response to your comments.

Comment:

Line 23-24: ‘This isn't a hypothesis. A hypothesis is an assumption, an idea that is proposed for the sake of argument so that it can be tested to see if it might be true. For example, in your study, a hypothesis would be something like: (i) frogs in urban environments will begin vocal activity earlier than those in non-urban areas; (ii) frog populations in urban areas will have shorter breeding seasons; (ii) urban noise will have negative effects on frog calling activity; and (iv) air and water temperature will positively influence frog calling activity.’

Response:

Thanks for your suggestion. We have deleted this sentence to avoid confusion,

Comment:

Line 25-26: ‘Suggestion: "which suggests a change induced by factors related to urbanization on reproduction phenology".’

Response:

We modified it in line 20-22.

Comment:

Line 29: ‘urban noise?’

Response:

We modified it in line 24.

Comment:

Line 30-31: ‘Ok, but what is the consequence of the results found?’

Response:

We appreciate your concern. We have added possible consequences of the results found. Please check line 25 – 27 of the revised version.

Comment:

Line 30: ‘frog breeding activities’

Response:

We modified it in line 28.

Comment:

Line 31-33: ‘How will this study help future research assess the impacts of urbanization on biodiversity and animal behavior?’

Response:

Thanks for your comment. We have deleted this sentence in the revised manuscript.

Comment:

Line 34: ‘This word is already in the title, change it to another one.’

Response:

Thanks for your concern. We have modified it in line 30 of the revised manuscript.

Comment:

Line 42-47: ‘Reference suggestion: Zaffaroni-Caorsi et al. 2022 - Effects of anthropogenic noise on anuran amphibians. https://doi.org/10.1080/09524622.2022.2070543’

Response:

We have included the suggested reference in the revised manuscript. Please check line 42.

Comment:

Line 98: ‘How will calling patterns be shaped? The response variable is the number of individuals in acoustic activity (Calling Index - CI), so the study is more related to breeding behavior (phenology).’

Response:

We modified it in line 92.

Comment:

Line 104: ‘Studied species and sites’

Response:

Thanks for your advice. We have modified it in the revised manuscript. Please check line 98.

Comment:

Line 110: ‘One reference is enough.’

Response:

We have modified it accordingly in line 104.

Comment:

Line 113: ‘Same comment as above.’

Response:

We have deleted additional references in line 107.

Comment:

Line 120: ‘Add a subtopic such as "Sampled Sites"’

Response:

We have added it in line 116.

Comment:

Line 127: ‘[63]’

Response:

We have deleted it in the revised manuscript.

Comment:

Line 164-166: ‘I saw in Table 3 that the call indexes (CI) of these species (Pelophylax nigromaculatus and Dryophytes japonicus) were included. However, I don't see much biological sense in adding these data to the model, since in the study hypothesis (not even mentioned in the introduction) there is nothing related to interspecific competition.’

Response:

Thanks for your comment. Both species are the most common sympatric species found with bullfrogs in South Korea, and they were included as just the sympatric species variables sharing habitat with bullfrog in the present study not to show the interspecific competition.

Comment:

Line 170: ‘What do you mean by "passing humans"?’

Response:

Thanks for your query. “Passing humans” means a person walking close to the recording equipment. However, we have modified the sentence to avoid confusions. Please check lines 165 – 166.

Comment:

Line 183-185: ‘Were human footsteps heard in the recordings? Do footsteps have a frequency higher than 90 Hz? This data seems to represent more the presence and absence of humans at that moment of the recording than "human noises".’

Response:

Thanks for your comment. Yes, we recorded it as presence and absence. Some sites (especially in type 3) had trails made of tree, and the sound of footsteps can be heard. The sounds were of more than 50Hz. As human noise was difficult to accurately estimate from a recorded file, we counted the presence and absence of human beings.

Comment:

Line 198: ‘ok, but I believe there may be some measurement error associated with this data, since it may be difficult to distinguish between two, three or four frogs calling at the same time, especially when they have similar call parameters and vocalize close to each other. Or not?’

Response:

We appreciate your comment. We agree with you that it is difficult to distinguish when many individuals call at the same time but not impossible. There are very small differences (dominant frequency, note duration, etc.) in the call of the individuals and we checked while watching spectrogram and listening to the sound source at the same time, so up to three frogs can be distinguished. You may also go through the following references for more information.

reference:

Zhang, F., Zhao, J., & Feng, A. S. (2017). Vocalizations of female frogs contain nonlinear characteristics and individual signatures. PloS one, 12(3), e0174815. https://doi.org/10.1371/journal.pone.0174815

Pettitt, B. A., Bourne, G. R., & Bee, M. A. (2013). Advertisement call variation in the golden rocket frog (Anomaloglossus beebei): evidence for individual distinctiveness. Ethology, 119(3), 244-256. https://doi.org/10.1111/eth.12058

Comment:

Line 243-244: ‘Add the average time of Type 3.’

Response:

We have added the average time of Type 3 in line 235.

Comment:

Line 260: ‘calling activity patterns’

Response:

We have modified it in line 25 of the revised manuscript.

Comment:

Line 267: ‘daily calling behavior’

Response:

We have modified it in line 258 – 259.

Comment:

Line 268-269: ‘This happens in South Korea, right? In tropical countries, the months (season) are quite different.’

Response:

Yes, this is the breeding seaso

---

## [Decision Letter · Decision Letter 1]

The influences of urbanization on breeding behavior of American bullfrog (Aquarana catesbeiana) in South Korea

PONE-D-24-38252R1

Dear Dr. Sung,

We’re pleased to inform you that your manuscript has been judged scientifically suitable for publication and will be formally accepted for publication once it meets all outstanding technical requirements.

Kind regards,

Daniel de Paiva Silva, Ph.D.

Academic Editor

PLOS ONE

Additional Editor Comments (optional):

Reviewers' comments:

Reviewer's Responses to Questions

**Comments to the Author**

Reviewer #1: All comments have been addressed

Reviewer #2: All comments have been addressed

2. Is the manuscript technically sound, and do the data support the conclusions?

Reviewer #1: Yes

Reviewer #2: Yes

3. Has the statistical analysis been performed appropriately and rigorously?

Reviewer #1: Yes

Reviewer #2: Yes

4. Have the authors made all data underlying the findings in their manuscript fully available?

Reviewer #1: Yes

Reviewer #2: Yes

5. Is the manuscript presented in an intelligible fashion and written in standard English?

Reviewer #1: Yes

Reviewer #2: Yes

Reviewer #1: I congratulate the authors for their thorough revision of the manuscript and for addressing all reviewer comments in a clear and appropriate manner. In my opinion, the manuscript now meets the standards for publication.

Reviewer #2: I believe that all the comments I made were taken into consideration and the corrections I pointed out were properly addressed. I suggested that it would be useful to incorporate a modeling tool to predict how the target species of the study could adapt to future climate change conditions and the consequent alterations in environmental (and water) temperatures. However, the authors chose to focus on the current conditions in the field to align with the primary objective of the study. I agree with this rationale and also, based on what was presented to me after my analysis, I find that the study meets the standards required by the journal.

**Do you want your identity to be public for this peer review?** For information about this choice, including consent withdrawal, please see our Privacy Policy

Reviewer #1: No

Reviewer #2: No

---

## [Editor Report · Acceptance letter]

PONE-D-24-38252R1

PLOS ONE

Dear Dr. Sung,

I'm pleased to inform you that your manuscript has been deemed suitable for publication in PLOS ONE. Congratulations! Your manuscript is now being handed over to our production team.

Kind regards,

on behalf of

Dr. Daniel de Paiva Silva

Academic Editor

PLOS ONE